# An Improved Version of the Generalized Laplacian Pyramid Algorithm for Pansharpening

Paolo Addesso [1,*], Rocco Restaino [1] and Gemine Vivone [2]

1 DIEM, University of Salerno, 84084 Fisciano, Italy; restaino@unisa.it
2 Institute of Methodologies for Environmental Analysis, CNR-IMAA, 85050 Tito Scalo, Italy; gemine.vivone@imaa.cnr.it
* Correspondence: paddesso@unisa.it

**Abstract:** The spatial resolution of multispectral data can be synthetically improved by exploiting the spatial content of a companion panchromatic image. This process, named pansharpening, is widely employed by data providers to augment the quality of images made available for many applications. The huge demand requires the utilization of efficient fusion algorithms that do not require specific training phases, but rather exploit physical considerations to combine the available data. For this reason, classical model-based approaches are still widely used in practice. We created and assessed a method for improving a widespread approach, based on the generalized Laplacian pyramid decomposition, by combining two different cost-effective upgrades: the estimation of the detail-extraction filter from data and the utilization of an improved injection scheme based on multilinear regression. The proposed method was compared with several existing efficient pansharpening algorithms, employing the most credited performance evaluation protocols. The capability of achieving optimal results in very different scenarios was demonstrated by employing data acquired by the IKONOS and WorldView-3 satellites.

**Keywords:** pansharpening; multispectral images; generalized laplacian pyramid; multilinear regression; filter estimation

## 1. Introduction

Pansharpening [1–5] has generated growing interest in the last years due to the numerous requests for accurate reproductions of the Earth surface, which pushed researchers to enhance the performance of algorithms based on remotely sensed data. Indeed, pansharpening represents a crucial step in the production of images aimed at visual interpretation in widely exploited software such as Google Earth and Bing Maps. Likewise, many other applications take advantage of this kind of fused data, for instance, agriculture (e.g., for crop type [6] and tree species [7] classification and for precision farming [8]), land cover change detection (e.g., for snow [9], forest [10] and urban [11] monitoring), archaeology [12] and even space mission data analysis [13].

Although the term is often used for a large set of combined data, the pansharpening process more exactly indicates the enhancement of a multispectral (MS) image through fusion with a higher resolution panchromatic (PAN) representing the same scene, as depicted in Figure 1. This setting allows one to obtain a very high quality final product, since the acquisitions can be collected almost contemporaneously from the same platform, thanks to the availability of the two required sensors on many operating satellites.

Several algorithms have been developed for completing the pansharpening process. They can be categorized in different ways according to their main features. In particular, an useful taxonomy that can guide the choice of the user distinguishes two main classes composed of classical and emerging approaches [5]. Essentially, the first group includes the methods which have been developed over the years, starting from the analysis of the physical processes underlying the acquisition of the signals involved. It includes

both component substitution (CS) approaches—for example those based on intensity-hue-saturation (IHS) [14,15], Gram–Schmidt [16,17] or principal component analysis [18,19] decompositions—and multiresolution analysis (MRA) methods, which use signal decompositions based on box filters, Laplacian pyramids [20–23], wavelets [24–26] and morphological filters [27–29]. Instead, the methods contained in the second group are more focused on the optimization of the fusion result, which aim at obtaining the best image quality through the application of more general estimation approaches. Techniques based on variational optimization (VO) approaches [30–32] and on machine learning (ML) [33–37] belong to this class.

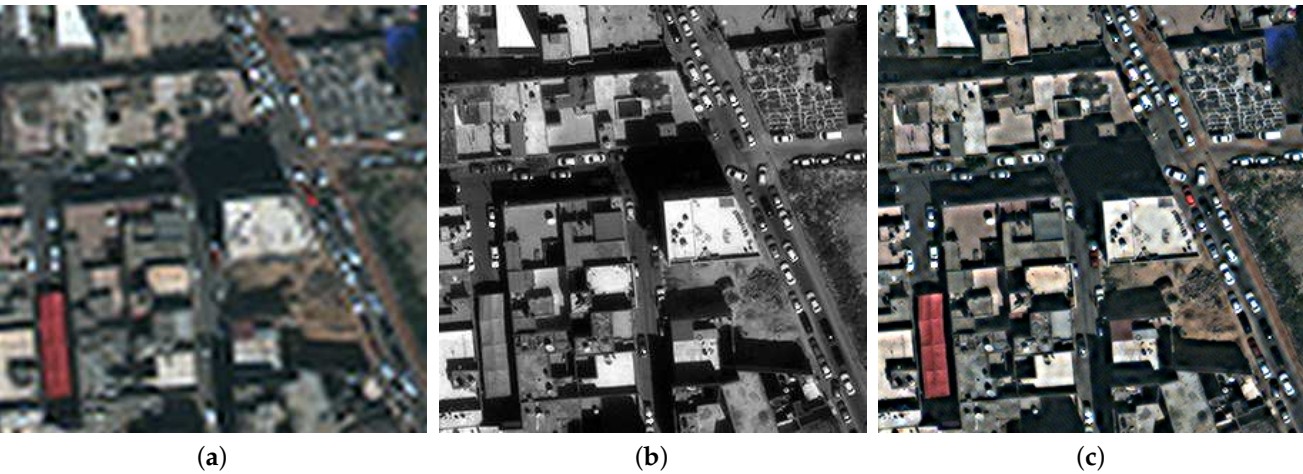

(**a**)　　　　　　　　　　　　(**b**)　　　　　　　　　　　　(**c**)

**Figure 1.** An example of pansharpening: (**a**) MS (interpolated) image; (**b**) PAN image; (**c**) fused image (namely, via the MBFE-BDSD-MLR method, presented in Section 5).

The literature shows several papers in which the results achievable through these approaches are compared [1,5,38–41]. As in many other fields, the recent development of more efficient computational approaches has constituted a major breakthrough in data fusion, making feasible the utilization of variational and ML-based methods. More in detail, in the last decade, pioneering works in ML category were compressive sensing and dictionary-based solutions, such as [31,42–46]. Subsequently, deep learning (DL) approaches became more and more popular in the remote sensing field [47], including pansharpening [33–36,48–55], and intimately related tasks such as super-resolution [56–59] and hyper-/multi-spectral data fusion [60,61]. The main issue of the above-mentioned ML approaches is the assumption of a training paradigm relying on a resolution downgrade process (e.g., Wald's protocol). More recently, different training paradigms, mainly based on multi-objective strategies, such as in [62,63], have been proposed to address such a problem.

On the other hand, a careful analysis of the literature testifies that the performance enhancements obtained through recent implementations of classical methods, such as those proposed in [64–66], or efficient implementations of VO, such as the one proposed in [30], lead to high quality pansharpened images that do not require extensive training phases.

For this reason, we focus in this paper on possible improvements of these efficient approaches, and in particular, we exploit the classical method scheme that is composed by two successive phases [1]: (i) the extraction of the details from a high resolution PAN image and (ii) the injection of those details into a low resolution MS image. We tackled the investigation of both phases, following the lines traced by the recent literature.

In more detail, we focus on an MRA approach based on the generalized Laplacian pyramid (GLP) [20] with a modulation transfer function (MTF)-matched filter. Namely, the details are extracted from the PAN image through a filter, whose amplitude response is matched to the MTF of the MS sensor [21]. This technique points toward obtaining the

most relevant data to the enhancement of the MS image spatial resolution, since it isolates the PAN information that was cut out off by the MS acquisition process. The cited reviews contain a vast list of pansharpening approaches using the MTF-shaped extraction filters. However, the exact form of the MS sensor's MTF is frequently unavailable in practice, due to the lack of accurate and updated on-board measurements of the actual response, which change over time due to the acquisition device aging [67]. Accordingly, good practice consists of estimating the actual shape of the detail-extraction filter directly from data [30]. If the response is significantly different across bands, it is advisable to estimate a different filter for each band [68].

Typically, the subsequent phase of detail injection is completed by adding the image extracted from the PAN image weighted by an injection coefficient matrix, which can in general contain a different entry for each pixel and each band. The values of the specific weights can be derived through physical considerations (as it happens for the HPM method [67,69]) or through mathematical optimization approaches starting from suitable criteria. The projective (or regression-based) injection model [21,70] belongs to the latter type, since it implements the minimum mean square error (MMSE) estimation approach. However, the described linear injection rule does not always represent the optimal choice, as it can also be argued from the studies that propose non-linear approaches, implemented through local methods [15,66,71] or non-linear networks [33,48]. In this work we elaborate on this thesis, while aiming to preserve the computational efficiency of classical methods. To that end, we adopted a slight generalization of the linear approach that consists of estimating the best polynomial approximation of the optimal relationship between the details extracted from the PAN image and those missing in the MS image. This approach allows one to estimate the optimal injection coefficients, according to the MMSE criterion, through a simple closed formula implementing a multilinear regression (MLR) scheme [29].

The original contribution of this study relies on the definition of a novel fusion architecture. The combination of the filter estimation and the MLR injection approach has been assessed and compared to several existing approaches. We show that with the conjunction of the two techniques, it is possible to obtain remarkable robustness against the diversity of the illuminated scenes, thereby enhancing a key feature of algorithms in practical applications. We tested the proposed method using two different real datasets, acquired from the IKONOS and the WorldView-3 sensors, which allowed us to evaluate the effective performance in different working scenarios. The quality of the final products has been assessed by exploiting both the reduced scale and the full scale assessment protocols [30]. The former allowed us to evaluate the performances of the algorithms in a controlled scenario, where the original MS images were used as references for the fusion of images that were purposely degraded to lower resolution. The latter constituted a realistic scenario in which the MS and PAN images were combined at the original resolution, in the absence of reference images.

In the next section, we describe the problem at hand, define the quantities used in the paper and provide an overview of the proposed approach. The following two sections are devoted to detailed descriptions of the chosen implementations of the two phases that compose the fusion process. In Section 5, we describe the simulation setting and report the outcomes of the experimental tests. The discussion of the results is in Section 6. Finally, conclusions are drawn in Section 7.

## 2. Problem Statement

Firstly, we introduce the mathematical notation that will be used in this paper. Bold uppercase, example $\mathbf{X}$, indicates an image. Accordingly, $\mathbf{P}$ is used to denote the panchromatic (PAN) image, which is an $L_r \times L_c$ two-dimensional array, where $L_r$ and $L_c$ are the numbers of rows and columns of the PAN image, respectively. Instead, the multispectral (MS) image is a three-dimensional array with dimensions $L_r/R \times L_c/R \times B$, where $B$ is the number of bands, and $R$ is the resolution ratio between the original MS and the PAN data; it is denoted by $\mathbf{M} = \{\mathbf{M}_b\}_{b=1,\dots,B}$, where $\mathbf{M}_b$ indicates the $b$-th spectral band.

The purpose of pansharpening is to identify a near-optimal procedure to obtain $L_r \times L_c \times B$ MS data—$\widehat{\mathbf{M}} = \{\widehat{\mathbf{M}}_b\}_{b=1,\dots,B}$—that is characterized by the same spectral resolution of the original MS image $\mathbf{M}$ and the same spatial resolution of the PAN image $\mathbf{P}$. We also define an $\widetilde{\mathbf{M}} = \{\widetilde{\mathbf{M}}_b\}_{b=1,\dots,B}$, the $L_r \times L_c \times B$ MS image obtained by solely upsampling $\mathbf{M}$ to the PAN scale.

The general formulation of a classical fusion process is given by the following expression [1]:

$$\widehat{\mathbf{M}}_b = \widetilde{\mathbf{M}}_b + F_b[\Delta\mathbf{P}_b], \ b = 1,\dots,B, \tag{1}$$

where:

- $\Delta\mathbf{P}_b$ are the PAN detail images, computed as $\Delta\mathbf{P}_b = \mathbf{P}_b - \mathbf{P}_b^L$, i.e., as the differences between the band by band equalized PAN image $\mathbf{P}_b$ (the equalization is performed as suggested in [27]) and its *low-pass filtered* version $\mathbf{P}_b^L$;
- $F_b[\cdot]$ are the functions (different for each band $b$) that inject the PAN details into each MS band.

According to (1), in classical methods each band is treated independently and an additive form is assumed for the injection procedure. The different techniques adopted to compute $\mathbf{P}_b^L$ (and hence the details $\Delta\mathbf{P}_b$) and $F_b[\cdot]$ identify the specific approaches. More specifically, if $\mathbf{P}_b^L$ is obtained by combining the channels of the interpolated MS image $\widetilde{\mathbf{M}}$, the method is said to belong to the CS class, whereas if $\mathbf{P}_b^L$ is obtained by extracting the low-pass part from the PAN image $\mathbf{P}$, the method is said to belong to the MRA class. This distinction is not purely formal, since the two classes can be shown to be characterized by very different visual and quantitative features [72].

Starting from this framework, in this work we propose an architecture, depicted in Figure 2, in which the main steps are the following ones.

(1) *Filter estimation*. The low-pass filtered PAN image $\mathbf{P}_b^L$ is obtained via an MRA scheme, where the low-pass filter impulse response $\mathbf{h}_b$ (that can be different for each band $b \in \{1,\cdots,B\}$) is estimated from the data by using semiblind deconvolution [30,68], as will be detailed in Section 3.
(2) *Multilinear regression-based injection*. The functions $F_b[\cdot]$ are supposed to have a non-linear form that can be approximated by a polynomial expansion of order $M$, whose coefficients are computed by using a multilinear regression (MLR) approach [29]. The complete description of this procedure will be provided in Section 4.

Indeed, our purpose was to evaluate the joint effect of both filter estimation and MLR-based injection on the final fused product. Therefore, in the following sections, we present in detail our proposal for implementing these two steps.

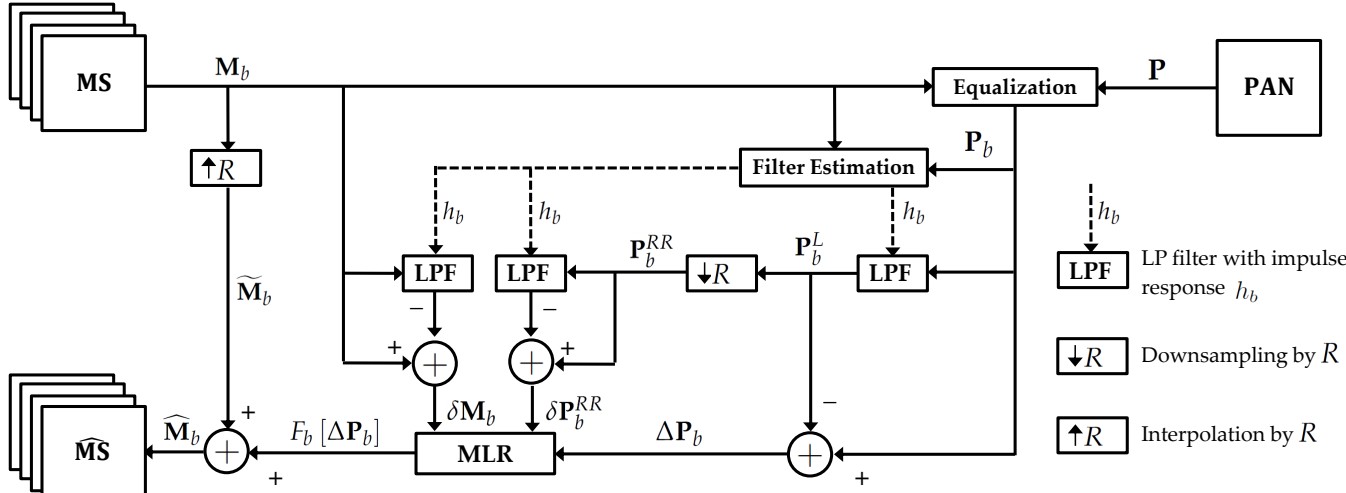

**Figure 2.** Block scheme of the pansharpening framework.

### 3. MRA via Filter Estimation

In this section we present two MRA approaches based on the estimation of the low-pass filter impulse response, namely, the filter estimation (FE) and the multi-band filter estimation (MBFE) algorithms, introduced in [30,68], respectively.

Both these algorithms rely on approaching the estimation problem in the following form. Let $\mathbf{x} \in \mathbb{R}^N$ be a single band lexicographic ordered image, where $N = L_r L_c$, and let $\mathbf{y} \in \mathbb{R}^N$ be its spatially degraded version, obtained via a low-pass filter (LPF) with the finite impulse response $\mathbf{h} \in \mathbb{R}^N$ and the addition of noise $\mathbf{n} \in \mathbb{R}^N$. In formulas, we can write the following equation:

$$\mathbf{y} = C(\mathbf{x})\mathbf{h} + \mathbf{n}, \tag{2}$$

where $C(\cdot)$ is an operator that generates a block circulant with circulant block (BCCB) matrix by suitably rearranging the entries of its argument such that the matrix product between $C(\mathbf{x}) \in \mathbb{R}^{N \times N}$ and $\mathbf{h}$ yields the convolution between the image $\mathbf{x}$ and the filter $\mathbf{h}$ [68,73].

In this setup, the filter estimation problem is addressed by solving the constrained minimization problem:

$$\min_{\mathbf{h}} \left\{ \|\mathbf{y} - C(\mathbf{x})\mathbf{h}\|^2 + \lambda \|\mathbf{h}\|^2 + \mu \left( \|C(\mathbf{d}_v)\mathbf{h}\|^2 + \|C(\mathbf{d}_h)\mathbf{h}\|^2 \right) \right\}$$
$$\text{subject to } \mathbf{h}^T \mathbf{1} = 1, \mathbf{h} \in \mathcal{H}, \tag{3}$$

where $\cdot^T$ is the transpose operator; $C(\mathbf{d}_v) \in \mathbb{R}^{N \times N}$ and $C(\mathbf{d}_h) \in \mathbb{R}^{N \times N}$ are BCCB matrices that are computed from the filters $\mathbf{d}_v$ and $\mathbf{d}_h$ and that perform the first-order finite difference in the vertical and horizontal directions, respectively; and $\mathbf{1}$ is a row vector of all ones. This formulation derives from the combination of different terms. The first one is the so-called *data-fitting term*, which is the main quantity to be optimized. Then, there are two regularization terms, introduced due to the ill-posedness of this estimation problem [74] that put some constraints on the resulting solution. The first regularization term (namely, $\|\mathbf{h}\|^2$) helps to obtain a *limited-energy* solution. On the other hand, the second regularization term (namely, $\|C(\mathbf{d}_v)\mathbf{h}\|^2 + \|C(\mathbf{d}_h)\mathbf{h}\|^2$) is useful to enforce a smooth solution. The weights $\lambda$ and $\mu$ aim to tune the importance of the regularization terms when finding the solution $\mathbf{h}$, which is subject to two constraints: it has to be normalized (i.e., $\mathbf{h}^T \mathbf{1} = 1$) and limited to a finite, non-empty and convex support, $\mathcal{H} \subset \mathbb{R}^N$. As stated in [30,68], the choice of the squared $\ell_2$ norm $\| \cdot \|^2$ allows one to solve the problem in (3) in a closed form. Indeed, the quadratic cost function within the minimization problem in (3) attains its (global) minimum when

$$\left[ C(\mathbf{x})^T C(\mathbf{x}) + \lambda \mathbf{I} + \mu C(\mathbf{d}_v)^T C(\mathbf{d}_v) + \mu C(\mathbf{d}_h)^T C(\mathbf{d}_h) \right] \mathbf{h} = C(\mathbf{x})^H \mathbf{y}, \tag{4}$$

where $\mathbf{I}$ is the identity matrix and $\cdot^H$ indicates the Hermitian transpose operator.

A computational efficient solution of (4) exists and can be explicitly written in the frequency domain as

$$\mathbf{h} = \mathcal{F}^{-1} \left\{ \circ \frac{\mathcal{F}\{\mathbf{x}\}^* \circ \mathcal{F}\{\mathbf{y}\}}{\mathcal{F}\{\mathbf{x}\}^* \circ \mathcal{F}\{\mathbf{x}\} + \lambda + \mu(\mathcal{F}\{\mathbf{d}_h\}^* \circ \mathcal{F}\{\mathbf{d}_h\} + \mathcal{F}\{\mathbf{d}_v\}^* \circ \mathcal{F}\{\mathbf{d}_v\})} \circ \right\}, \tag{5}$$

where the $\circ$ symbol indicates pointwise (entry-by-entry) multiplication and division operations; $\mathcal{F}\{\cdot\}$ and $\mathcal{F}^{-1}\{\cdot\}$ are the 2-D discrete Fourier transform operator and its inverse, respectively; and $(\cdot)^*$ denotes the complex conjugate. Indeed, due to their BCCB structure, the matrices $C(\mathbf{x})$, $C(\mathbf{d}_h)$ and $C(\mathbf{d}_v)$, can be diagonalized by the 2D discrete Fourier transform matrix, leading to a computational cost dominated by the number of operations required to perform the fast Fourier transform (FFT) transform, i.e., $O(N \log N)$. Finally, in order to fully profit from the FFT, which assumes a periodic boundary structure of the image, a preprocessing step aimed at smoothing out the unavoidable discontinuities

present in the real-world images is needed. The adopted solution relies on blurring the image borders, as suggested in the image processing literature [75].

In the following, we briefly describe the two most promising and effective approaches to complete the filter estimation task for pansharpening, i.e.,:

- The filter estimation (FE) method [30], which estimates a single filter for all the MS channels;
- The multi-band filter estimation (MBFE) algorithm [68], which overcomes the main limitation of the FE by estimating a filter for each band.

### 3.1. FE Algorithm

This approach consists of estimating a single filter; i.e., for all spectral bands $b \in \{1, \cdots, B\}$, $\mathbf{h}_b$ is equal to the same estimated filter, say $\mathbf{h}_P$. The $P$ subscript refers to the implementation of the FE algorithm that exploits the relationship between the PAN image $\mathbf{p} \in \mathbb{R}^N$ and an equivalent PAN image $\mathbf{p}_e \in \mathbb{R}^N$ generated by projecting the MS image into the PAN domain via the formula

$$\mathbf{p}_e = \mathbf{w}^{aug}\widetilde{\mathbf{M}}^{aug}. \tag{6}$$

In (6), $\widetilde{\mathbf{M}}^{aug} = \left[\widetilde{\mathbf{m}}_1^T, \cdots, \widetilde{\mathbf{m}}_b^T, \cdots, \widetilde{\mathbf{m}}_B^T, \mathbf{1}^T\right]^T$ is obtained by stacking the lexicographic ordered single band images $\{\widetilde{\mathbf{m}}_b\}_{b=1,\dots,B}$ and the all-ones row vector $\mathbf{1}$, and $\mathbf{w}^{aug} = \left[\mathbf{w}^T, w_0\right]^T$ is composed by the vector $\mathbf{w} = [w_1, \cdots, w_b, \cdots, w_B]$, whose elements are the *weights* measuring the overlap between the PAN image and each spectral band, and $w_0$ is a bias coefficient.

The algorithm is based on the following two interdependent alternating steps, starting from an initial estimate for the filter $\mathbf{h}_P$.

- *Estimation of the weights* $\mathbf{w}^{aug}$. This step consists of imposing the equality between the image $\mathbf{p}_e$ defined in (6) and a low-pass filtered version of $\mathbf{P}_b$ computed via the current estimate of $\mathbf{h}_P$. Therefore, the estimate of the weights $\mathbf{w}^{aug}$ is easily found via a classic multivariate regression framework.
- *Estimation of the filter* $\mathbf{h}_P$. This estimate is found by using (5) in which $\mathbf{p}_e$ plays the role of the blurred and degraded image $\mathbf{y}$ and $\mathbf{p}$ plays the role of the matrix $\mathbf{x}$. The resulting filter is finally normalized (in order to have a unitary gain) and the values outside a given window are set to zero.

In order to help a fast convergence of the iterative algorithm (usually in a couple of iterations), the Starck and Murtagh low-pass filter [76] (used in pansharpening in the popular "à trous" algorithm) is chosen as initial guess for the filter $\mathbf{h}_P$. Moreover, the maximum number of iterations is fixed to 10, in order to ensure that the algorithm stops.

### 3.2. MBFE Algorithm

An effective algorithm aimed at estimating a specific degradation filter $\mathbf{h}_b$ for each $b \in \{1, \cdots, B\}$ is the MBFE method [68]. The low coherence between some bands of the MS image and the PAN image prevents satisfying performance by estimating $\mathbf{h}_b$ directly from the PAN image [30]. This problem is solved by generating an initial estimate of $\widehat{\mathbf{M}}$ (say it $\widehat{\mathbf{M}}^0 = \{\widehat{\mathbf{M}}_b^0\}_{b=1,\dots,B}$) that can be used as an approximation of the ground-truth (GT) for the filter estimation task, ensuring higher coherence between the high and the low resolution MS images.

Natural candidates for estimating $\widehat{\mathbf{M}}^0$ are the CS-based methods. Indeed, they are able to generate a fused product that completely retains the PAN spatial details [30], which are key for an accurate estimation of the filters. Therefore, for each band $b \in \{1, \cdots, B\}$, the estimation of $\mathbf{h}_b$ is performed by using (5) in which $\widetilde{\mathbf{m}}_b$ plays the role of the blurred and degraded image $\mathbf{y}$ and $\widehat{\mathbf{m}}_b^0$ (i.e., the lexicographic ordered version of $\widehat{\mathbf{M}}_b^0$) plays the role of the matrix $\mathbf{x}$. Among the CS algorithms, the band-dependent spatial-detail (BDSD) technique [77] and the Gram–Schmidt adaptive (GSA) method [17] have been proved to

generate initial guess images $\widehat{\mathbf{M}}^0$ that are well-suited for use in MBFE [68]. Therefore, in the following, we will consider two MBFE variants: the MBFE BDSD and the MBFE GSA.

## 4. MLR-Based Injection

In this section we briefly present the MLR-based injection scheme that is the second pillar of the proposed pansharpening architecture. This scheme is a natural extension of the following classical approaches, in which $F_b[\cdot]$ is linear.

- *CBD Injection Scheme.* In the context-based decision (CBD) injection model, for each channel $b$, the details of the PAN image are multiplied by a scalar coefficient, namely,

$$F_b[\Delta\mathbf{P}_b] = g_b\Delta\mathbf{P}_b. \tag{7}$$

The *injection coefficients* $g_b, \forall b \in \{1, \cdots, B\}$ are computed by the regression of the $b$-th MS channel on the PAN images. It is worth noting that this scheme is also used in other pansharpening algorithms, such as the aforementioned GSA.

- *HPM Injection Scheme.* The high-pass modulation (HPM) injection scheme relies on the pointwise multiplication of the PAN details by a *coefficient matrix* $\mathbf{G}_b$, according to

$$F_b[\Delta\mathbf{P}_b] = \mathbf{G}_b \circ \Delta\mathbf{P}_b. \tag{8}$$

Additionally, in this case, other pansharpening algorithms use this scheme, such as the Brovey transform (BT), which is a classic multiplicative scheme belonging to the CS family [78].

On the contrary, the proposed scheme employs a non-linear injection function $F_b[\cdot]$ in (1) that is approximated via a polynomial expansion of order $M$ around zero, i.e.,

$$F_b[\Delta\mathbf{P}_b] = \sum_{m=0}^{M} g_{b,m}(\Delta\mathbf{P}_b)^m. \tag{9}$$

This formulation is linear with respect to the coefficients $\{g_{b,m}\}_{m=0,\dots,M}$; therefore, it is possible to use the MLR framework [79] that is the multidimensional extension of the classic ordinary least square method. More in detail, we should estimate the coefficients $\{g_{b,m}\}_{m=0,\dots,M}$ by solving the problem

$$\Delta\widehat{\mathbf{M}}_b = \sum_{m=0}^{M} g_{b,m}(\Delta\mathbf{P}_b)^m + \mathbf{R}_b \tag{10}$$

where $\Delta\widehat{\mathbf{M}}_b = \widehat{\mathbf{M}}_b - \widetilde{\mathbf{M}}_b$ are the details of the target MS image and the optimal (in the least-squares sense) coefficients $\{g_{b,m}\}_{m=0,\dots,M}$ minimize the Frobenius norm of the residuals $\mathbf{R}_b$. Unfortunately, it is impossible to compute the details of the MS image, because this approach would require the knowledge of $\widehat{\mathbf{M}}_b$—that is, the image to estimate. Therefore, the solution is to solve the reduced resolution companion problem

$$\delta\mathbf{M}_b = \sum_{m=0}^{M} g_{b,m}^{RR}(\delta\mathbf{P}_b^{RR})^m + \mathbf{R}_b^{RR}, \tag{11}$$

defined in terms of the corresponding reduced resolution versions, indicated by the superscript *RR*. More specifically,

$$\begin{aligned} \delta\mathbf{M}_b &= \mathbf{M}_b - h_b * \mathbf{M}_b, \\ \delta\mathbf{P}_b^{RR} &= \mathbf{P}_b^{RR} - h_b * \mathbf{P}_b^{RR}, \end{aligned} \tag{12}$$

where $\mathbf{P}_b^{RR}$ is the downsampled version of $\mathbf{P}_b^L$, defined in Section 2 (see also Figure 2). Finally, according to the findings of [29], we use $M = 2$, which shows a good trade-off between the complexity of the model and its performance.

## 5. Experimental Results

The crucial phase of this study was constituted by the experimental tests that have allowed us to verify the suitability of the proposed technique. Due to the above-mentioned lack of a reference image, the assessment of fusion algorithms had to be performed in two distinguished steps, involving the evaluation of the performance at both reduced scale and full scale [1]. The former consisted of reproducing the fusion problem at a lower resolution through the appropriate degradation of the available images. In particular, the choice of a scaling factor equal to the resolution ratio $R$ allows one to downsize the PAN image at the resolution of the original MS image, which can thus be used as the fusion target. In this case, some indexes can be used for the evaluation of the image quality. On the other hand, changing the working resolution does not represent an ideal solution for two main reasons. The information concerning the illuminated area is somewhat different at the two scales, due to a substantial reduction of perceivable details. Moreover, the image processing algorithms employed for coarsening the available images can only approximate the sensor acquisition, leading to significant deviations form the actual operating scenarios. For this reason, the analysis of the algorithms' behavior at the original resolution is mandatory, especially if the reliability of the applicable quality measures is highly questionable.

We utilized two different datasets for the assessment of the proposed technique, comparing it to the MS image interpolation using a polynomial kernel with 23 coefficients (EXP) and many classical approaches, i.e., non-linear IHS (NLIHS) [15], Gram–Schmidt (GS) [16], Gram–Schmidt adaptive (GSA) [17], band-dependent spatial-detail (BDSD) [77], smoothing filter-based intensity modulation (SFIM) [69], additive à trous wavelet transform (ATWT) [26] and the pyramidal decomposition scheme using morphological filters based on half gradient (MF-HG) [80]. Obviously, we included in the comparison the baseline methods that constitute the starting point of the proposed approach, namely, the GLP using MTF-matched filter [21] with both multiplicative (HPM) [26] and regression-based (CBD) [70] injection models. Moreover, we considered the more recent versions of the MTF-GLP approaches that include the filter estimation procedure, based on the estimation of either a single filter (FE) [30], or a different filter for each band (MBFE) [68]. Analogously, we report the results related to the sole introduction of the MLR injection scheme [29] within the baseline methods.

### 5.1. Datasets

The datasets were selected to cover the most typical settings encountered in the practice. For this reason, two different cases have been considered: the fusion of a PAN image with a 4-band MS image and the fusion of a PAN image with an 8-band MS image. Moreover, the two datasets refer to different scenarios, one constituted by a mountainous, partly vegetated area, and one representing an urban zone.

The *China dataset* was also used in the assessment of the classical methods presented in [1] and is composed by images collected by the IKONOS sensor over the Sichuan region in China (see the images on the left in Figure 3). The MS image had four channels (blue, green, red and near infra-red (NIR)) and the spatial sampling interval (SSI) was 1.2 m. The IKONOS resolution ratio between the MS and the PAN image was $R = 4$ and the radiometric resolution was 11 bits. A cut of size $300 \times 300$ pixels of the original MS image was employed in this work as the ground-truth (GT) for the reduced resolution assessment.

The *Tripoli dataset* was acquired by the WorldView-3 satellite and was used to test the capability of the proposed method for the enhancement of an MS with a larger number of bands (coastal, blue, green, yellow, red, red edge, NIR1 and NIR2). The employed imagery included a PAN image of size $1024 \times 2048$ pixels and an MS image of size $256 \times 256$ pixels, which was also in this case $R = 4$. The MS WorldView (WV)-3 sensor is characterized by a 1.2 m SSI and the radiometric resolution is 11 bits. The *Tripoli dataset* was used both at the original scale for performing the full resolution assessment and at a lower scale, obtained by degrading the original images by a factor $R = 4$.

**China**　　　　　　　　　**Tripoli**

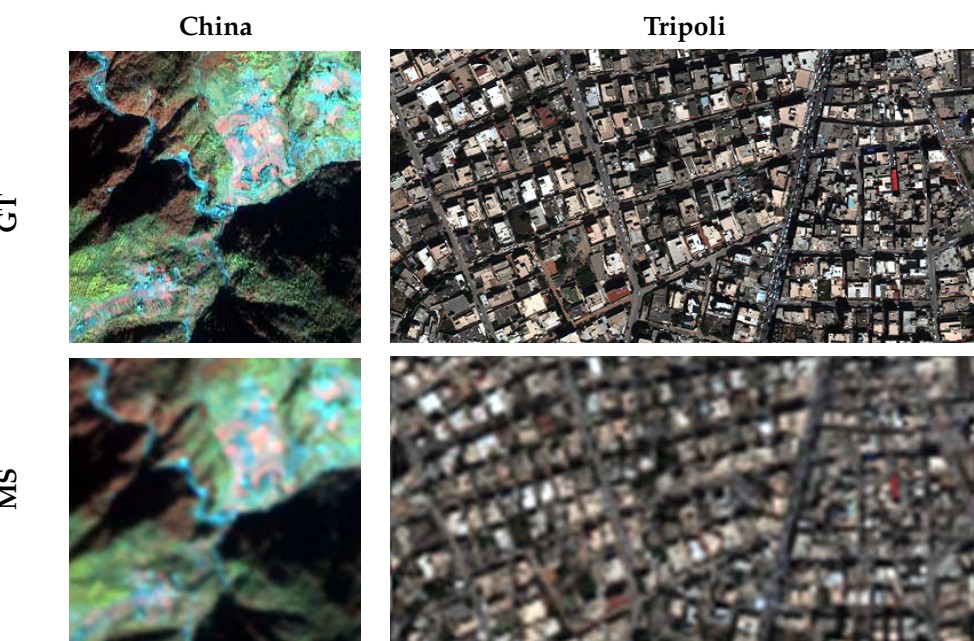

**Figure 3.** Reduced resolution datasets: *China dataset* (on the left), *Tripoli dataset* (on the right). The first row reports the original MS image (used as the ground-truth) and the second row reports the degraded MS image, upsampled to the ground-truth size.

### 5.2. Reduced Resolution Validation

The reduced resolution (RR) assessment represents a very crucial phase in the evaluation of algorithm performance, since the availability of the reference image allows one to accurately evaluate the quality of the final product. The method consists of using the original MS image as the target product that has to be obtained by combining the degraded versions of the same MS image and of the original PAN. The procedure for completing such experiments has been formalized through Wald's protocol [81], which is based on both the consistency and the synthesis properties. While the latter points at specifying the characteristics of the fused image, the former requires that the fused high resolution MS image obtains the low resolution MS image once properly degraded. This means that the degradation systems should reproduce the overall acquisition process that yields the real images. Accordingly, the amplitude frequency responses of the MS degradation filters are matched to the MTFs of the MS sensor, while an ideal filter is employed to decimate the PAN image [21].

The quality of the fused products can thus be assessed through several indexes that have been developed. We adopted four widespread measures: the well-known *peak signal-to-noise ratio* (PSNR); the *relative dimensionless global error in synthesis* (ERGAS) [82] that is a normalized version of the root mean square error (RMSE); the *spectral angle mapper* (SAM) [83] that quantifies the spectral distortion as the mean angle between the fused and reference pixel vectors; the $Q2^n$ [84,85] that extends the universal image quality index (UIQI) [86] to multi-channel images.

The main results of the RR assessment are summarized in Table 1, where the comparison of the proposed technique with both the baseline methods and the other cited pansharpening approaches is shown.

In order to give some additional insight about data fusion performance, we also present some closeups for the two RR data in Figures 4 and 5, focusing only on the GLP details extraction scheme, we show several $Q2^n$ maps for the Tripoli dataset in Figure 6.

**Table 1.** Reduced resolution assessment: the first row contains the reference value for each index. The best results among the different versions of the MTF-GLP approaches are in boldface; the second best are underlined.

| Algorithm | | China | | | | Tripoli | | | |
|---|---|---|---|---|---|---|---|---|---|
| | | *PSNR* | *ERGAS* | *SAM* | *Q4* | *PSNR* | *ERGAS* | *SAM* | *Q8* |
| Reference | | ∞ | 0 | 0 | 1 | ∞ | 0 | 0 | 1 |
| EXP | | 36.012 | 3.8736 | 4.4268 | 0.7389 | 20.408 | 9.2964 | 8.6679 | 0.6197 |
| NL-IHS | | 38.115 | 3.2280 | 4.0268 | 0.7968 | 22.685 | 7.2531 | 8.8094 | 0.7719 |
| GS | | 39.833 | 2.8310 | 3.5399 | 0.8475 | 23.554 | 6.5690 | 8.3871 | 0.8166 |
| GSA | | 40.711 | 2.5829 | 3.0053 | 0.8756 | 26.597 | 4.7677 | 7.5888 | 0.9220 |
| BDSD | | 41.003 | 2.4361 | 2.9272 | 0.8884 | 25.645 | 5.2490 | 8.1877 | 0.9133 |
| SFIM | | 39.904 | 2.6076 | 3.2165 | 0.8731 | 24.381 | 5.9649 | 8.1231 | 0.8585 |
| ATWT | | 40.270 | 2.5483 | 3.0916 | 0.8793 | 24.769 | 5.7169 | 7.8750 | 0.8750 |
| MF-HG | | 40.085 | 2.6472 | 3.1053 | 0.8669 | 24.880 | 5.6423 | 7.9807 | 0.8874 |
| HPM | MBFE BDSD | 41.010 | 2.4904 | 2.9926 | 0.8824 | 25.181 | 5.4896 | 8.0262 | 0.8899 |
| | MBFE GSA | 40.973 | 2.5100 | 3.0095 | 0.8816 | 25.234 | 5.4580 | 7.9807 | 0.8887 |
| | FE | 41.011 | 2.4913 | 2.9871 | 0.8820 | 25.222 | 5.4672 | 7.9785 | 0.8882 |
| | GLP | 40.993 | 2.4834 | 2.9985 | 0.8825 | 25.147 | 5.5124 | 7.9880 | 0.8839 |
| CBD | MBFE BDSD | 40.914 | 2.5495 | 3.0055 | 0.8776 | 26.298 | 4.9096 | 7.9474 | 0.9224 |
| | MBFE GSA | 40.866 | 2.5754 | 3.0209 | 0.8762 | 26.641 | 4.7440 | 7.6183 | 0.9248 |
| | FE | 40.914 | 2.5495 | 3.0036 | 0.8773 | 26.627 | 4.7530 | 7.6086 | 0.9247 |
| | GLP | 40.936 | 2.5311 | 2.9709 | 0.8784 | 26.614 | 4.7546 | <u>7.5353</u> | 0.9211 |
| MLR | MBFE BDSD | <u>41.179</u> | 2.4149 | 2.9249 | **0.8885** | 26.305 | 4.9091 | 7.9526 | 0.9235 |
| | MBFE GSA | 41.158 | 2.4304 | <u>2.9179</u> | 0.8877 | **26.687** | **4.7251** | 7.5565 | **0.9263** |
| | FE | 41.177 | <u>2.4145</u> | 2.9255 | <u>0.8882</u> | 26.670 | 4.7358 | 7.5441 | <u>0.9262</u> |
| | GLP | **41.220** | **2.4006** | **2.8589** | 0.8876 | 26.660 | <u>4.7341</u> | **7.4512** | 0.9222 |

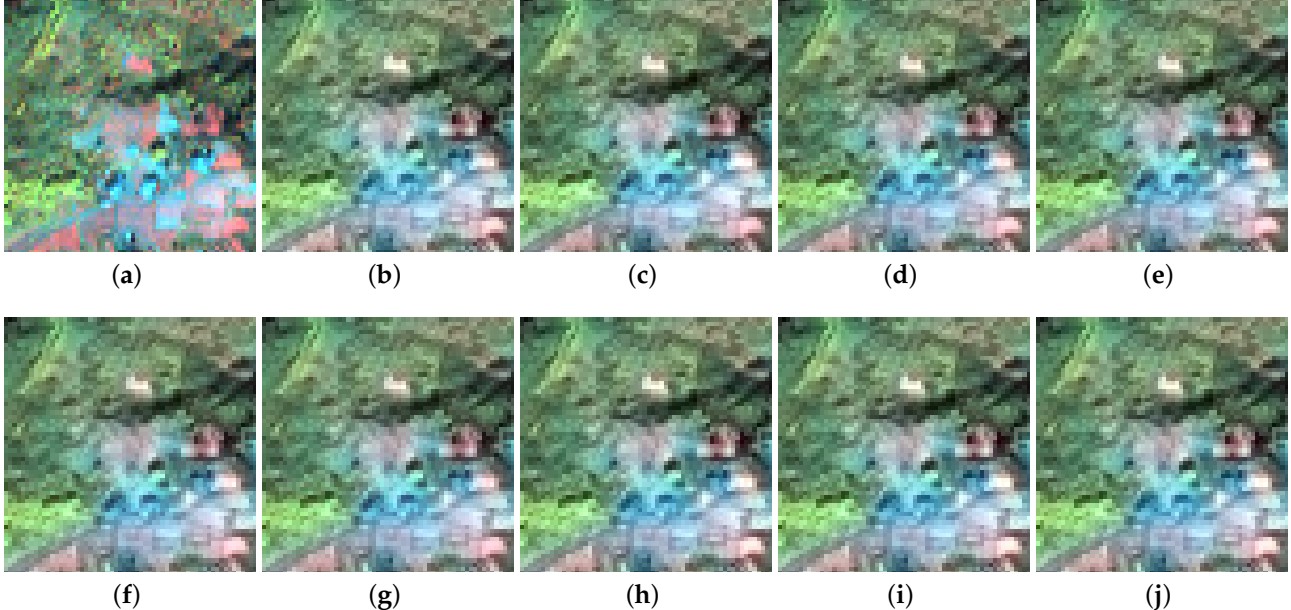

(**a**)    (**b**)    (**c**)    (**d**)    (**e**)

(**f**)    (**g**)    (**h**)    (**i**)    (**j**)

**Figure 4.** *Cont.*

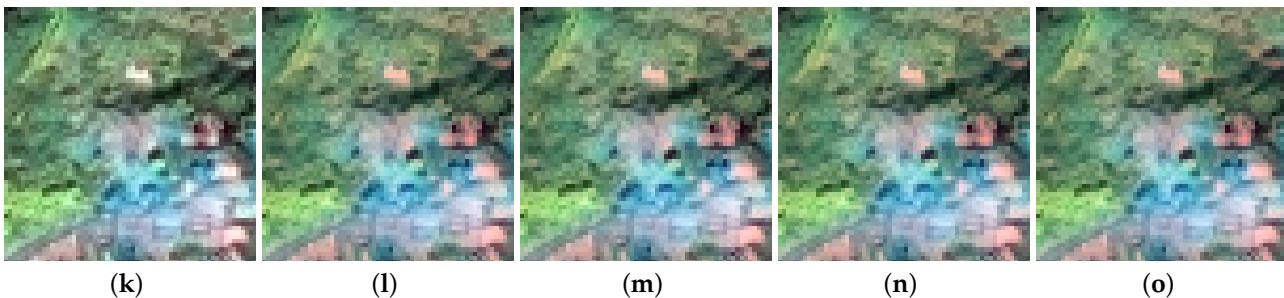

**Figure 4.** Close-ups of the fused results using the reduced resolution China dataset: (**a**) GT; (**b**) GSA; (**c**) MF-HG; (**d**) MBFE-BDSD-HPM; (**e**) MBFE-GSA-HPM; (**f**) FE-HPM; (**g**) GLP-HPM; (**h**) MBFE-BDSD-CBD; (**i**) MBFE-GSA-CBD; (**j**) FE-CBD; (**k**) GLP-CBD; (**l**) MBFE-BDSD-MLR; (**m**) MBFE-GSA-MLR; (**n**) FE-MLR; (**o**) GLP-MLR.

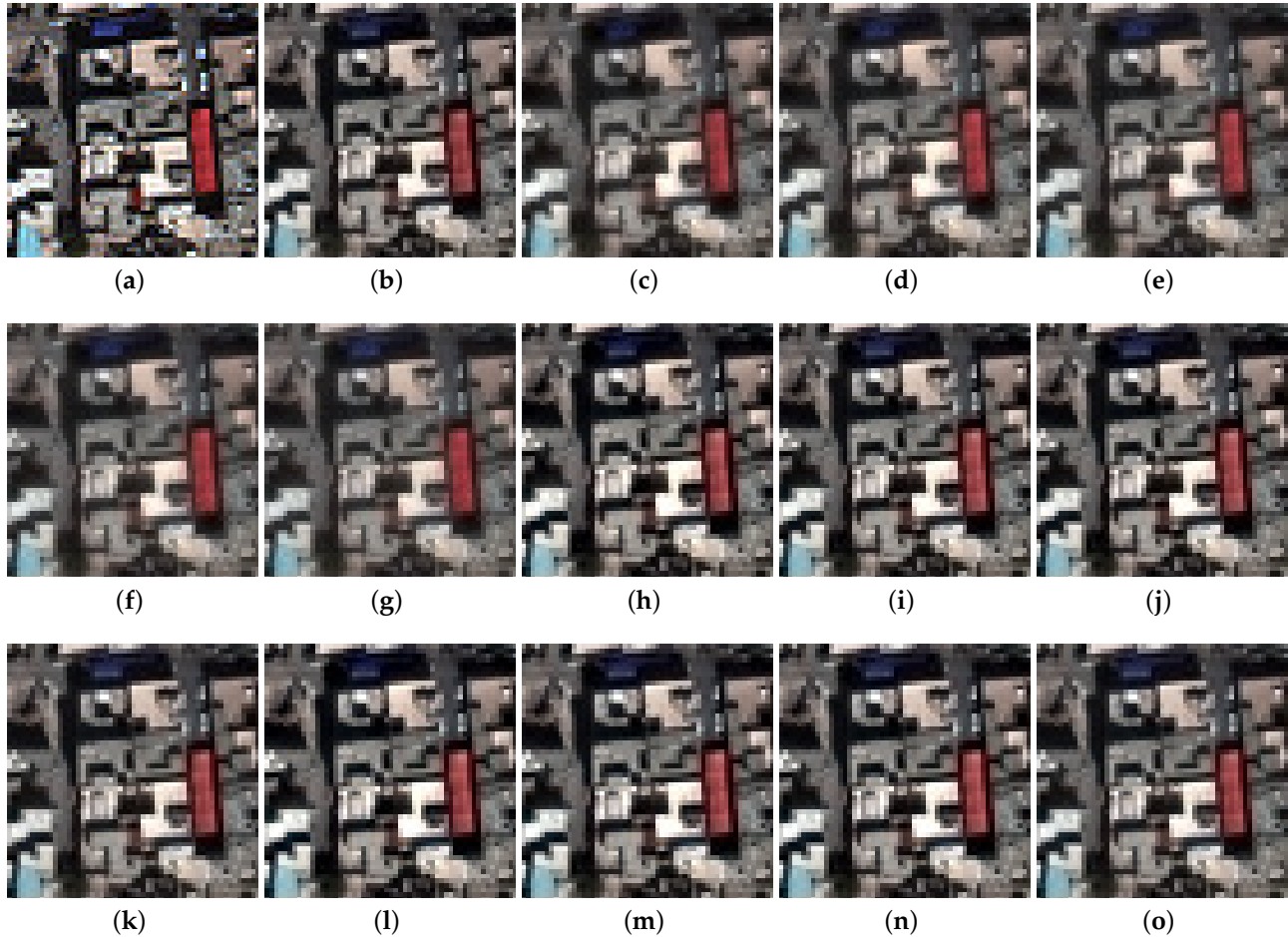

**Figure 5.** Close-ups of the fused results using the reduced resolution Tripoli dataset: (**a**) GT; (**b**) GSA; (**c**) MF-HG; (**d**) MBFE-BDSD-HPM; (**e**) MBFE-GSA-HPM; (**f**) FE-HPM; (**g**) GLP-HPM; (**h**) MBFE-BDSD-CBD; (**i**) MBFE-GSA-CBD; (**j**) FE-CBD; (**k**) GLP-CBD; (**l**) MBFE-BDSD-MLR; (**m**) MBFE-GSA-MLR; (**n**) FE-MLR; (**o**) GLP-MLR.

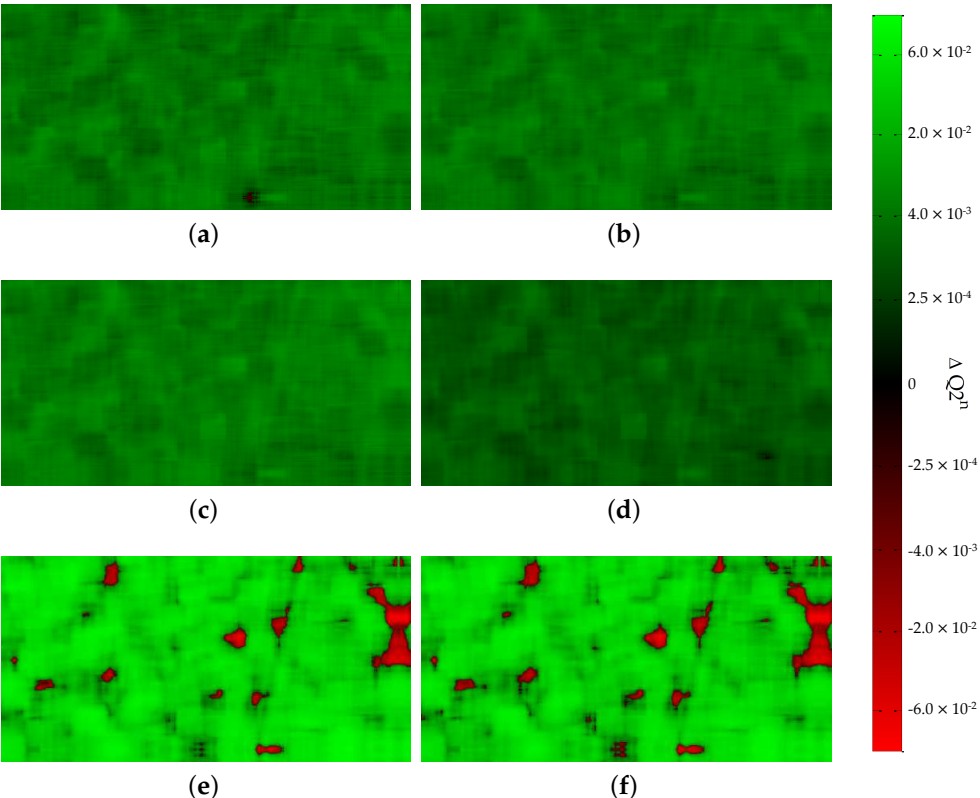

**Figure 6.** Tripoli dataset (RR): differences between the $Q2^n$ maps computed for the best algorithm, i.e., MBFE-GSA-MLR, and the baseline methods, i.e., (**a**) GSA, (**b**) GLP-MLR, (**c**) GLP-CBD, (**d**) MBFE-GSA-CBD, (**e**) GLP-HPM and (**f**) MBFE-GSA-HPM. Green values: better results obtained by MBFE-GSA-MLR; red values: better results obtained by the other algorithm.

### 5.3. Full Resolution Validation

The full resolution (FR) assessment allowed us to analyze the behavior of the algorithms at their effective working scale. In particular, the Tripoli dataset contains images with high resolution details, namely, with physical dimensions very similar to the SSI of the sensors. Accordingly, it can be properly used for this second investigation phase, in which the visual analysis assumes a central role. In fact, all the available quality indexes cannot be considered totally reliable because they assess the final product without a reference image.

For this reason, the quantitative evaluation is typically performed by measuring the coherence of the pansharpened product with the original available images. In particular, one assesses the spectral similarity of the fused image and the low resolution MS image and the correspondence between the PAN and MS spatial details at the original and enhanced resolutions [87,88].

The quality with no reference (QNR) [88] index is the best known measure adopting this rationale. It is composed by a spectral index that measures the relationships among the MS channels and a spatial index that quantifies the quantity and the appropriateness of the spatial details present in each band. Several other quality indexes have been proposed in the literature for the full resolution assessment [23,89–92]. We adopted here the hybrid QNR (HQNR) [93] that combines the use of the QNR spatial index $D_S$ and of the spectral index $D_\lambda$ proposed in [89], thereby providing appreciable soundness and computational efficiency [5].

Table 2 reports the values of the HQNR indexes computed by applying the considered pansharpening algorithms to the Tripoli dataset.

**Table 2.** Full resolution assessment on the Tripoli dataset: the first row contains the reference for each indicator. Best results for the two tested injection schemes (HPM and CBD) are in boldface; the second best are underlined.

| | Algorithm | $D_s$ | $D_\lambda$ | $HQNR$ |
|---|---|---|---|---|
| | **Reference** | 0 | 0 | 1 |
| | **EXP** | 0.0717 | 0.0317 | 0.8989 |
| | **NL-IHS** | 0.0602 | 0.0795 | 0.8651 |
| | **GS** | 0.0767 | 0.0536 | 0.8738 |
| | **GSA** | 0.0775 | 0.0377 | 0.8877 |
| | **BDSD** | 0.0755 | 0.1399 | 0.7952 |
| | **SFIM** | 0.0657 | 0.0216 | 0.9141 |
| | **ATWT** | 0.0654 | 0.0174 | 0.9183 |
| | **MF-HG** | 0.0591 | 0.0183 | 0.9236 |
| **HPM** | **MBFE BDSD** | 0.0650 | 0.0181 | 0.9181 |
| | **MBFE GSA** | 0.0667 | 0.0188 | 0.9157 |
| | **FE** | 0.0679 | 0.0179 | 0.9155 |
| | **GLP** | 0.0684 | 0.0183 | 0.9146 |
| **CBD** | **MBFE BDSD** | 0.0637 | 0.0169 | <u>0.9205</u> |
| | **MBFE GSA** | 0.0661 | 0.0178 | 0.9173 |
| | **FE** | 0.0674 | <u>0.0167</u> | 0.9171 |
| | **GLP** | 0.0697 | 0.0171 | 0.9144 |
| **MLR** | **MBFE BDSD** | **0.0590** | 0.0196 | **0.9226** |
| | **MBFE GSA** | <u>0.0626</u> | 0.0199 | 0.9188 |
| | **FE** | 0.0634 | 0.0188 | 0.9190 |
| | **GLP** | 0.0698 | **0.0158** | 0.9156 |

## 6. Discussion

The reduced resolution and the full resolution assessment protocols allowed us to provide a clear illustration of the results achievable through the proposed pansharpening scheme. The key considerations that can be derived from these two complementary phases are detailed in the following sections.

### 6.1. Reduced Resolution

The most evident property is the high performance achieved by the proposed approach in both the tests. This behavior stood out on our datasets, presenting complementary features of the illuminated scene. Indeed, as it can be noticed by examining most of the compared algorithms, it is difficult to find algorithms that were characterized by optimal performance in both the scenarios. Additionally, the baseline methods, namely, the implementations of the GLP approaches exploiting the HPM and the CBD injection schemes, were not immune to this performance tradeoff, due to the large presence of high resolution details in the Tripoli datasets, whose counterparts are the large homogeneous zones in the China dataset.

In both the cases, the best results in terms of the most comprehensive index, namely, the $Q2^n$, were obtained by the methods that utilize the multi-band filter estimation. In particular, the MBFE-BDSD-MLR approach achieved the highest value for the China dataset, and the MBFE-GSA-MLR produced the best image for the Tripoli dataset. An important remark regards the single filter estimation approach (FE-MLR) that showed remarkable robustness, since it achieved results very similar to the best methods; this was partly due to the shape of the MTFs of the various bands, which are characterized by almost equal gains at the Nyquist frequency. Moreover, one can note that the improvement of the final product quality implied by the filter estimation procedure is always in terms of spectral quality of the image, as demonstrated by the higher values of the SAM index with respect to the

GLP-MLR. In fact, the MLR coefficient estimation points to optimizing the detail injection scheme for each specific channel, without taking into account the spectral coherence of the final product. Naturally, this issue is made worse by the multiband approach, which estimates a different filter for each band, causing a more significant spectral unbalance in the pansharpened image. Nevertheless, this issue is largely compensated by the greater ability of injecting the most useful spatial information contained in the PAN image.

The suitability of the proposed approaches is also testified by the closeups shown in Figures 4 and 5 that highlight the capability of producing images with accurate spatial reproduction of the details, without excessively sacrificing the chromatic fidelity of the objects presents in the scene. The performance analysis can be eased by evaluating the algorithms in pairs, as we show in Figure 6, where the MBFE-GSA-MLR is compared, in terms of $Q2^n$ map, to six other methods based on the GLP detail-extraction scheme. The green pixels represent the zones of the images in which the MBFE-GSA-MLR achieved higher quality index scores with respect to the competitors, and the red pixels highlight the opposite. The proposed method achieved almost uniform performance improvements with respect to the approaches compared in panels (a)–(d). More specifically, figures (b) and (d) show that the MBFE-GSA-MLR is significantly superior to the algorithms that implement either the MLR injection scheme (GLP-MLR) or the MBFE technique (MBFE-GSA-CBD), thereby motivating the joint use of the two methods. The uniformity of the green pixels demonstrate that the performance increase was not due to the improvement of a specific zone of the image, but rather to a more precise evaluation of the best extraction filter and of a specific formula for the data combination. Panels (e) and (f) illustrate a more diversified result that is a consequence of an alternative injection scheme. In fact, the HPM modulated the PAN details point-wise, thereby achieving a very different final product. However, the higher overall quality of the MBFE-GSA-MLR approach can be easily argued by the larger extent of the green zones, which are also characterized by high color saturation, indicating a significant improvement of the $Q2^n$ value.

*6.2. Full Resolution*

The results corroborate the analysis carried out at reduced resolution, showing that the approaches using an estimated filter for the extraction and a multilinear regression for the injection obtained the best overall results. In particular, the use of a specific index $D_S$ assessing the spatial quality of the images allows one to confirm the deduction that the main improvements were obtained in terms of a more faithful reproduction of the geometric information.

Further information can be derived from the visual analysis of the pansharpened products. We present in Figure 7 the injected details, namely, the differences between the final products $\widehat{\mathbf{M}}$ and the upsampled image $\widetilde{\mathbf{M}}$. Panels (m) and (n) immediately stand out for the richness and intelligibility of the representation, which testify the accurate reproduction of the object borders detectable at the highest scale. Moreover, although Table 2 confirms that the proposed methods resulted in slightly worse spectral quality of the final products, the homogeneity of the detailed images demonstrates that the particulars were not excessively boosted in any specific zone or band, as happened for the HPM-based schemes.

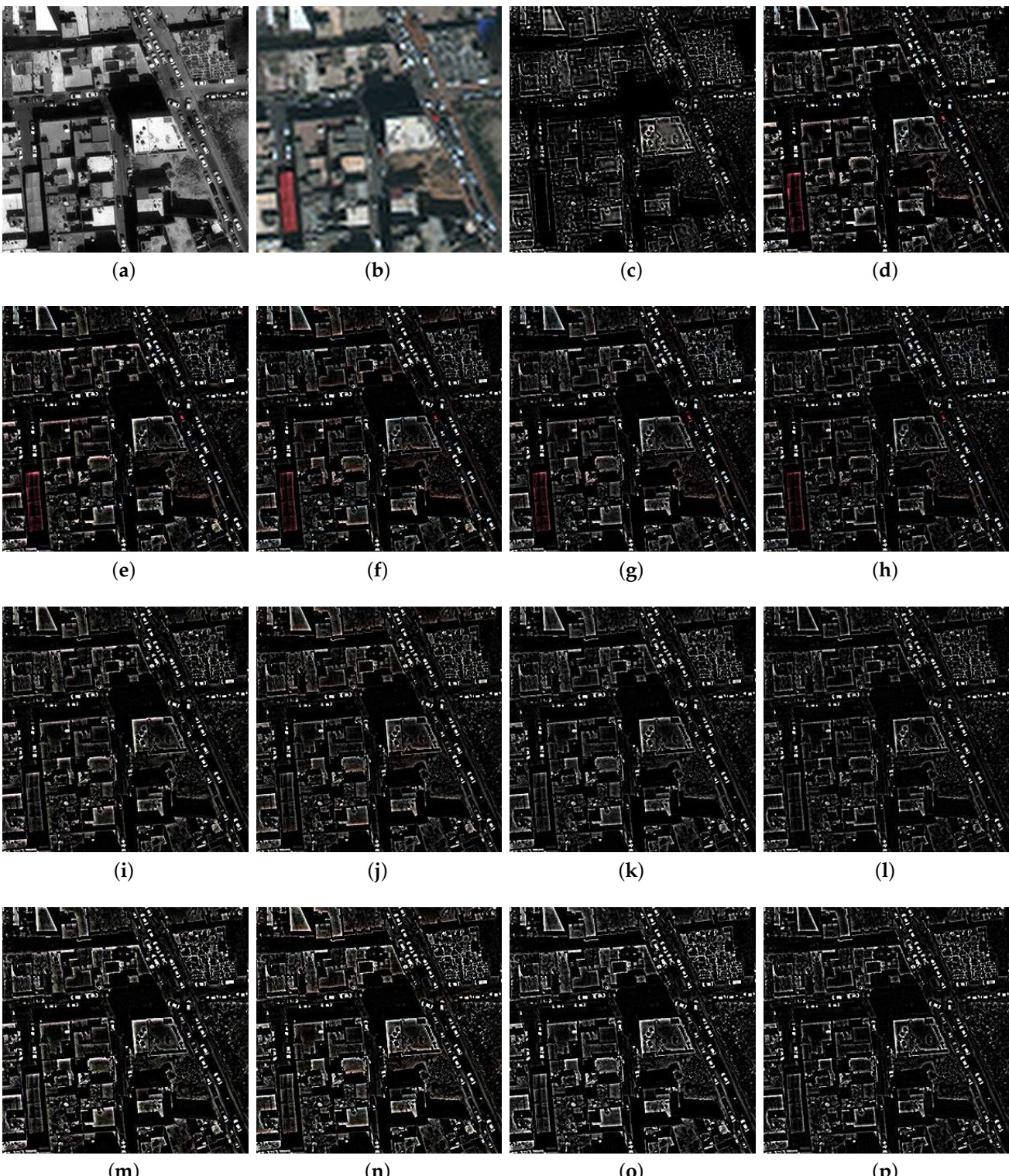

**Figure 7.** Close-ups of the details of the fused results using the full resolution Tripoli dataset: (**a**) PAN; (**b**) EXP; (**c**) details for GSA; (**d**) details for MF-HG; (**e**) details for MBFE-BDSD-HPM; (**f**) details for MBFE-GSA-HPM; (**g**) details for FE-HPM; (**h**) details for GLP-HPM; (**i**) details for MBFE-BDSD-CBD; (**j**) details for MBFE-GSA-CBD; (**k**) details for FE-CBD; (**l**) details for GLP-CBD; (**m**) details for MBFE-BDSD-MLR; (**n**) details for MBFE-GSA-MLR; (**o**) details for FE-MLR; (**p**) details for GLP-MLR.

*6.3. Computational Analysis*

The analysis of the computational complexity of the proposed approach is finally reported in Table 3, which contains the times required by an Intel®Core™I7 3.2GHz processor to complete the fusion process. Almost all the approaches exploiting a multiresolution decomposition of the images required perceptibly more computational effort, since the considered images are quite large. A further increase occurred for the filter estimation procedure, whose effort is proportional to the number of impulse responses to be estimated. Accordingly, the multiband (MBFE) approach took twice as much computational time as the baseline GLP approach in the case of eight bands, though the additional effort required by the single filter (FE) method is almost negligible. In any case, the main point is that the proposed approach strictly preserves the feasibility of the classical methods, thereby representing a viable technique for processing a large amount of data.

**Table 3.** Computational times (in seconds) required for the datasets used.

| | Algorithm | China (RR) | Tripoli (RR) | Tripoli (FR) |
|---|---|---|---|---|
| | **NL-IHS** | 1.104 | 4.049 | 74.64 |
| | **GS** | 0.0409 | 0.127 | 1.79 |
| | **GSA** | 0.0970 | 0.253 | 2.73 |
| | **BDSD** | 0.115 | 0.207 | 1.51 |
| | **SFIM** | 0.0223 | 0.102 | 1.35 |
| | **ATWT** | 0.117 | 0.570 | 8.76 |
| | **MF-HG** | 0.135 | 0.303 | 3.62 |
| **HPM** | **MBFE BDSD** | 0.298 | 0.929 | 14.72 |
| | **MBFE GSA** | 0.283 | 0.979 | 15.40 |
| | **FE** | 0.147 | 0.427 | 6.70 |
| | **GLP** | 0.123 | 0.412 | 5.56 |
| **CBD** | **MBFE BDSD** | 0.314 | 0.949 | 14.60 |
| | **MBFE GSA** | 0.307 | 1.013 | 15.47 |
| | **FE** | 0.170 | 0.429 | 6.65 |
| | **GLP** | 0.123 | 0.411 | 5.45 |
| **MLR** | **MBFE BDSD** | 0.298 | 2.058 | 15.10 |
| | **MBFE GSA** | 0.306 | 2.132 | 15.88 |
| | **FE** | 0.164 | 1.127 | 7.08 |
| | **GLP** | 0.154 | 1.103 | 7.07 |

## 7. Conclusions

In this work, a step forward has been made with respect to existing techniques for the efficient combinations of multispectral and panchromatic images acquired by the same satellite. The usefulness of pansharpened data for many applications demands the capacity of providers to efficiently perform the fusion process, and thus most algorithms still resort to physical models to ease their adaptation to specific datasets.

In this class of approaches, the generalized Laplacian pyramid has emerged as the most widespread method, since it combines the accurate reproduction of the acquisition process characteristics with high computational efficiency. However, some improvements that do not result in an excessive computational burden are conceivable. More specifically, in this work we validated the joint use of a filter estimation procedure, which allows one to easily adapt the shape of the detail-extraction filters to the specific imagery, and of a polynomial combination function, which allows one to more properly inject the PAN information.

The effectiveness of the proposed scheme has been tested on two different datasets, which are characterized by unalike features of the illuminated scene and have been acquired by different sensors. The most important quality of the designed approach is the capacity to achieve the best performance among the tested methods in both the scenarios, differently from all the existing techniques. Among the possible implementations of the proposed

approach, it has also been highlighted in this study that the estimation of a single filter for all the multispectral image channels allows one to obtain a still more efficient algorithm, without significantly sacrificing the overall quality of the final product.

Finally, future studies and developments will be devoted to extending the proposed architecture to hyperspectral sharpening, due to the great interest for the related applications [94,95], and the fusion of thermal data [96].

**Author Contributions:** Conceptualization, P.A., R.R. and G.V.; methodology, P.A., R.R. and G.V.; software, P.A., R.R. and G.V.; validation, P.A., R.R. and G.V.; formal analysis, P.A., R.R. and G.V.; investigation, P.A.; data curation, G.V.; writing—original draft preparation, R.R. and P.A.; writing—review and editing, G.V.; visualization, P.A. and R.R.; supervision, G.V. All authors have read and agreed to the published version of the manuscript.

**Funding:** This research received no external funding.

**Data Availability Statement:** The data presented in this study are available on request from the corresponding author. The data are not publicly available since have been removed from the original url.

**Conflicts of Interest:** The authors declare no conflict of interest.

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
