# Peer review of "An Improved Version of the Generalized Laplacian Pyramid Algorithm for Pansharpening"

_remotesensing, doi:10.3390/rs13173386_

Round 1

Reviewer 1 Report

This paper proposed a pansharpening method based on modified GLP. The experiment demonstrates the effectiveness. To improve the manuscript, the following questions need to be considered:

  1. Deep learning pansharpening methods should be mentioned.
  2. A flowchart of the algorithm should be given.
  3. The running time should be given in Table 1.
  4. PSNR index should be calculated for assessment.

Reviewer 2 Report

The manuscript entitled "An improved version of the generalized Laplacian pyramid algorithm for pansharpening" adopted a slight generalization of the linear approach that consists in estimating the best polynomial approximation of the optimal relationship between the details extracted from the PAN image and those missing in the MS image.

The authors demonstrated a good knowledge of the problem and of the related literature. The introduction section includes all significant related information, sufficiently highlighting the importance of the topic and the novelty of this work. All parts of the methodology are described in a comprehensive manner. The results are valuable and are properly discussed.

Nonetheless, still some reviews should be performed.

Line 99-104: Is this description really necessary? 

According to Instructions for Authors, Research manuscripts should comprise: Introduction, Materials and Methods, Results, Discussion, and Conclusions.

Author Response

Please, see the attached file

Reviewer 3 Report

This is a paper concerning pansharpening, i.e. using a high resolution panchromatic image to acquire a high resolution hyperspectral image.

First of all I want the Authors to look at this publication:

Hyperspectral Image Super-resolution via Deep Spatio-spectral Convolutional Neural Networks
Jin-Fan Hu, Ting-Zhu Huang, Liang-Jian Deng, Tai-Xiang Jiang, Gemine Vivone, Jocelyn Chanussot
(https://arxiv.org/abs/2005.14400)
How can their effort be compared with this paper?

Alsp what are the applications of pan-sharpenning a Hyperspectral Image. What are you trying to improve apart from the visual appearence?

This paper lacks applications and proper motivation for keeping the reader interested. While I have no doubpt their effort is quite hard. I do not think this work is yet suitable for immediate publication.

Please kindly show a low resolution of a Hyperspectral Image and a High resolution panchromatic image and blend them together showing it in the very start of the paper. 

Also please show clearly the innovation compared to previous works. It is not clearly shown.

Author Response

Please, see the attached file

Reviewer 4 Report

Please see the attached report.

Author Response

Please, see the attached file

Reviewer 5 Report

Manuscript ID: remotesensing-1309940

Manuscript Title: An improved version of the generalized Laplacian pyramid algorithm for pansharpening 

The paper is very well-established and the results are almost satisfactory. I have a couple of issues as follows:

1) Page 5, Line 181: What did you set for the range of w's?It has to be mentioned.

2) It is a good idea to test the performance of the method by changing the M to higher values (the order of polynomial expansion). For example, add results of M=3 and M=4.

3) The processing time for each method (including M=3 and M=4 mentioned in item (2) above) must be a part of experimental results in a separate table.

Author Response

Please, see the attached file

Round 2

Reviewer 2 Report

The authors have successfully addressed all comments. Also, the authors did a good job on replying to other reviewer's comments. Overall, the quality of this manuscript is significantly improved from the previous version. 

Reviewer 3 Report

Ok by me with a note: Although I spoke about hyperspectral please consider that there are deep learning works on the specific topic for multispectral images kindly have a look for future works.

Reviewer 5 Report

No more comments.